# Transient Autoreactive PF4 and Antiphospholipid Antibodies in COVID-19 Vaccine Recipients

**DOI:** 10.3390/vaccines11121851

**Published:** 2023-12-14

**Authors:** Matthijs P. Raadsen, Chantal Visser, A. H. Ayesha Lavell, Anita A. G. A. van de Munckhof, Jonathan M. Coutinho, Moniek P. M. de Maat, Corine H. GeurtsvanKessel, Marije K. Bomers, Bart L. Haagmans, Eric C. M. van Gorp, Leendert Porcelijn, Marieke J. H. A. Kruip

**Affiliations:** 1Department of Viroscience, Erasmus MC, University Medical Center Rotterdam, 3000 CA Rotterdam, The Netherlands; m.p.raadsen@erasmusmc.nl (M.P.R.); c.geurtsvankessel@erasmusmc.nl (C.H.G.); b.haagmans@erasmusmc.nl (B.L.H.); e.vangorp@erasmusmc.nl (E.C.M.v.G.); 2Department of Hematology, Erasmus MC, University Medical Center Rotterdam, 3000 CA Rotterdam, The Netherlands; c.visser@erasmusmc.nl (C.V.); m.demaat@erasmusmc.nl (M.P.M.d.M.); 3Department of Internal Medicine, Amsterdam UMC Location Vrije Universiteit Amsterdam, Boelelaan 1117, 1081 HV Amsterdam, The Netherlands; a.lavell@amsterdamumc.nl (A.H.A.L.); m.bomers@amsterdamumc.nl (M.K.B.); 4Amsterdam Institute for Infection & Immunity, Meibergdreef 9, 1105 AZ Amsterdam, The Netherlands; 5Department of Neurology, Amsterdam UMC Location University of Amsterdam, Meibergdreef 9, 1105 AZ Amsterdam, The Netherlands; a.a.vandemunckhof@amsterdamumc.nl (A.A.G.A.v.d.M.); j.coutinho@amsterdamumc.nl (J.M.C.); 6Department of Immunohematology Diagnostics, Sanquin Diagnostic Services, Plesmanlaan 125, 1066 CX Amsterdam, The Netherlands; l.porcelijn@sanquin.nl

**Keywords:** autoantibodies, COVID-19 vaccines, platelet factor 4, thrombosis, thrombocytopenia

## Abstract

Vaccine-induced immune thrombotic thrombocytopenia (VITT) is a rare autoimmune condition associated with recombinant adenovirus (rAV)-based COVID-19 vaccines. It is thought to arise from autoantibodies targeting platelet factor 4 (aPF4), triggered by vaccine-induced inflammation and the formation of neo-antigenic complexes between PF4 and the rAV vector. To investigate the specific induction of aPF4 by rAV-based vaccines, we examined sera from rAV vaccine recipients (AZD1222, AD26.COV2.S) and messenger RNA (mRNA) based (mRNA-1273, BNT162b2) COVID-19 vaccine recipients. We compared the antibody fold change (FC) for aPF4 and for antiphospholipid antibodies (aPL) of rAV to mRNA vaccine recipients. We combined two biobanks of Dutch healthcare workers and matched rAV-vaccinated individuals to mRNA-vaccinated controls, based on age, sex and prior history of COVID-19 (AZD1222: 37, Ad26.COV2.S: 35, mRNA-1273: 47, BNT162b2: 26). We found no significant differences in aPF4 FCs after the first (0.99 vs. 1.08, mean difference (MD) = −0.11 (95% CI −0.23 to 0.057)) and second doses of AZD1222 (0.99 vs. 1.10, MD = −0.11 (95% CI −0.31 to 0.10)) and after a single dose of Ad26.COV2.S compared to mRNA-based vaccines (1.01 vs. 0.99, MD = 0.026 (95% CI −0.13 to 0.18)). The mean FCs for the aPL in rAV-based vaccine recipients were similar to those in mRNA-based vaccines. No correlation was observed between post-vaccination aPF4 levels and vaccine type (mean aPF difference −0.070 (95% CI −0.14 to 0.002) mRNA vs. rAV). In summary, our study indicates that rAV and mRNA-based COVID-19 vaccines do not substantially elevate aPF4 levels in healthy individuals.

## 1. Introduction

In response to the COVID-19 pandemic, caused by the severe acute respiratory syndrome-associated coronavirus-2 (SARS-CoV-2), the first licensed vaccines against a human coronavirus were developed. Multiple clinical trials worldwide have proven the safety and efficacy of all licensed COVID-19 vaccines [1]. However, most high-income countries have favored messenger RNA (mRNA)-based vaccines for mass-immunization programs because the recombinant adenovirus vectored (rAV)-based vaccines had been associated with rare thrombotic adverse events, known as vaccine-induced immune thrombotic thrombocytopenia (VITT) [2,3,4,5,6,7,8].

In VITT, autoantibodies bind to platelet factor 4 (PF4), after which interaction with Fcy receptor IIA causes platelet activation and aggregation, with subsequent thrombosis and thrombocytopenia [4,5,7,8,9,10,11]. PF4 is a cationic tetramer and a chemokine [12] that is secreted during platelet activation binds to negatively charged surfaces (polyanions) and is believed to play a role in innate immunity [13]. VITT has clinical and serological similarities to heparin-induced thrombocytopenia (thrombosis) (HIT(T)), an autoimmune disorder caused by autoantibodies against PF4 after exposure to heparin, which is also a polyanionic molecule that can form large complexes with PF4 in vitro [12,14,15]. In vitro evidence suggests that PF4-heparin complexes act as a neoantigen that sensitizes B-cells to PF4, resulting in the formation of PF4 antibodies [16].

PF4 antibodies can also form in the absence of a specific antigen, in rare instances resulting in ‘spontaneous HITT’ [17,18]. In these cases, clinical signs of HITT typically develop shortly after an infection or surgical procedure. Recently, PF4 antibodies were found in over 95% of unvaccinated hospitalized COVID-19 patients, independent of prior heparin treatment [19]. Even in heparinized patients, HITT and PF4 antibodies are most commonly found shortly after surgery [20,21,22,23]. B cells expressing anti-PF4 were frequently detected in neonatal cord blood, stimulated in vitro without polyanions or PF4, indicating this autoantibody is part of the repertoire of naïve B cells [24]. The association between HITT and a recent infectious or inflammatory episode is a feature common to other autoimmune diseases [25]. Tissue damage, infection, and inflammation can also elicit antibodies against a range of other autoantigens, as has been demonstrated for several viral and bacterial infections, and recently in COVID-19 [26,27,28]. These antibodies are the result of a temporary lifting of immune tolerance due to inflammation and bystander activation of marginal zone B-cells that produce polyreactive antibodies as part of a physiological innate immune response [29,30]. Overall, data is suggesting that PF4 autoantibodies and HITT can arise through mechanisms similar to other autoimmune disorders, without requiring sensitization by a specific neoantigen.

Likewise, it is unclear whether the formation of PF4 neoantigen complexes is a prerequisite to the pathogenesis of VITT after administration of a rAV-based COVID-19 vaccine. Other factors potentially contributing to the risk of VITT have been proposed, including excess levels of host cell proteins and Ethylenediaminetetraacetic acid (EDTA) found in some batches of AZD1222 [4,31]. Understanding the relative contributions of rAV-induced neoantigen formation versus vaccine impurities is crucial for the further development of rAV-based vaccines, as the former may be inherent to the vector while the latter can be addressed during vaccine production.

To determine whether rAV-based vaccines specifically induce PF4-specific autoantibodies, this study combined two biorepositories of vaccinated health-care workers (HCWs) who had received rAV-based vaccines (AZD1222, Ad26.COV2.S) or mRNA vaccines (mRNA-1273, BNT162b2) and had their autoantibody response measured. Our analysis encompassed the assessment of the fold change (FC) of aPF and aPL after vaccination over baseline, the correlation between aPF and aPL levels, the percentage of HCWs with seroconversion, and the correlation between vaccine type and post-vaccination aPF levels. By doing so, we aimed to establish whether rAV-based vaccines uniquely induce PF4-specific autoantibodies, independently of non-VITT related antiphospholipid (aPL) antibodies, as a proxy for a nonspecific, polyreactive response commonly seen after viral infections and vaccines [32,33].

## 2. Materials and Methods

### 2.1. Study Population

A cohort of presumed healthy individuals of 18 years and older who received a primary series of AZD1222, AD26.COV2.S, mRNA-1273 or BNT162b2 in early 2021 was generated by combining two prospective biorepository studies of HCWs, described elsewhere [34,35]. The biorepository of the Erasmus Medical Center (EMC) included HCWs between February and June 2021 (cohort 1, *n* = 547 samples). Amsterdam University Medical Centers (AUMC) included HCWs between January and May 2021 (cohort 2, *n* = 284 samples). HCWs were excluded if they did not consent to the reuse of their data. Written informed consent was obtained from all HCWs and the studies were approved by the local ethical review boards of the respective institutions (MEC-2020-0264, NL73478.029.20).

HCWs were vaccinated according to Dutch national COVID-19 immunization guidelines in effect at the time of inclusion. These guidelines specified that AZD1222 should be avoided in HCWs below 60 years of age due to the observed association between AZD1222 and thrombosis, from April 2021 onwards [36]. The local institutional policy was to prioritize vaccination for frontline workers who had frequent direct contact with vulnerable patients, followed by support and research staff. Except for this guidance, the allocation of different vaccine types was based on the availability of the vaccines on the day of vaccination. Of note, HCWs were aware of the type they would receive before their scheduled vaccination appointment, which means we cannot exclude self-selection. Vaccines were dosed in accordance with the manufacturer’s instructions at the time of their introduction, which specified two doses, except for Ad26.COV.2.S, which is licensed as a single-dose vaccine [37].

### 2.2. Datasets

In this study two datasets were used: one for longitudinal (LO) analysis and another for cross-sectional (CS) analysis of T3. To determine the induction of PF4 and aPL antibodies after COVID-19 vaccination, the LO dataset was selected from the two combined biorepositories. HCWs were selected if serum samples collected before vaccination (T1) and after the first (T2) and/or second vaccine dose (T3) were available. Individuals who were incompletely vaccinated or with a heterologous regimen (mixing multiple vaccine types) were excluded. We used a matching approach, in which HCWs vaccinated with a rAV vaccine were matched to those vaccinated with an mRNA vaccine, based on age, sex, and prior COVID-19 infection status, with a preferential selection of male pairs to obtain a more representative study population (Figure 1). We used the Mahalanobis distance, which is a measure of divergence between two individuals concerning the covariates, with a value of 0 being a perfect match. Matched pairs of interest were selected for analysis based on the following criteria, in order of priority: type of rAV vaccine used (1:1 AZD1222 to AD26.COV2.S ratio), sex (1:1 female to male ratio) and exactness of the match (lowest Mahalanobis distance). For the CS dataset, we selected all HCWs with a serum sample collected after the last available vaccine dose in the series. Datasets were defined prior to obtaining the autoantibody results.

We further explored correlations between aPF4 levels after vaccination and the type of vaccine, using the CS dataset. All HCWs with a serum sample collected after the last vaccine dose in the series were selected for the CS dataset. Datasets were defined prior to obtaining the autoantibody results.

#### 2.2.1. PF4 Antibody ELISA

Anti-PF antibody levels were measured using an in-house anti-PF4 IgG ELISA was used. A 96-wells plate (Thermo Fisher Scientific, Waltham, MA, USA) was coated with 100 µL PF4 (3 µg/mL, ChromaTec, Greifswald, Germany) for 1.5 h at 37 °C, and washed with 0.05% Tween in PBS. After washing, the participant’s serum was added and incubated for one hour at room temperature. After incubation plates were washed and GaH-HRP IgG (IgG-HRP + TRIS-washing buffer/BSA 0.2%; Jackson ImmunoResearch, Westgrove, PA, USA) was added and incubated for another hour. The plate was washed and OPD substrate (2.5 mL PCbuffer + 2 mg O-Phenylenediamine; Sigma-Aldrich, Zwijndrecht, The Netherlands + 0.001% H_2_O_2_) was added. The plate was incubated for five minutes at room temperature. The reaction was stopped with H_2_SO_4_. Absorption was measured at 492 nm. Results were expressed in optical density (OD) ratios. The PF4 ELISA is deemed positive with OD values above 2.0 and dubious with OD values between 1.0 and 2.0.

#### 2.2.2. PF4-Dependent Platelet Activation Assay (PIPAA)

In the case of an OD above 1.0, a PIPAA was performed, as previously described by Greinacher et al. [38] with slight modifications. Whole blood was obtained from four healthy donors. Platelets were purified in the presence of adenine citrate dextrose (ACD) solution A. Platelet-rich plasma was resuspended in ACD and Apyrase (Sigma–Aldrich, Zwijndrecht, The Netherlands) and, after centrifugation for seven minutes at 650 g and washing with washing buffer, resuspended in Tyrode’s buffer. We incubated 75 µL platelet suspension in a microtiter plate with 20 µL patient serum and 10 µL PF4. The microtiter plate was placed on a magnetic stirrer (50 min, 500–600 rpm) with two steel spheres in each well. Every five minutes transparency of the suspension was assessed using an indirect light source. Furthermore, inhibition of platelet activation with 5 µL FcɣRIIa-blocking monoclonal antibody IV.3 (STEMCELL Technologies, Vancouver, BC, Canada) was performed. Platelet activation was established if aggregates formed within 45 min or less. Out of four donors, at least three donors had to show activation for the PIPAA to be scored positive. As a positive control, we included pooled human sera from patients diagnosed with HIT, with strong reactive heparin/PF4-antibodies and platelet activation occurring within 10 min in the PIPAA. As a negative control, we used inert sera from healthy donors with blood group AB. The PIPAA is considered positive if platelet activation occurs in the presence of PF4. Activation was measured by time elapsed until platelet activation.

#### 2.2.3. Antiphospholipid Antibody ELISA

Anti-beta2-Glycoprotein I (aβ2 GP) total immunoglobulin (including IgG, IgM, and IgA) and anti-Cardiolipin (aCL) immunoglobulin M (IgM) and IgG were quantitatively determined using diagnostic ELISA kits (KA1273 from Abnova—Taipei, Taiwan, and ORG515 from Orgentec—Mainz, Germany). Kits were used according to the manufacturer’s instructions and internally validated for sensitivity, specificity, and linear detection range using a panel of positive and negative control sera.

### 2.3. Endpoints

The main endpoint of the study was the mean difference (MD) of the fold change (FC) in PF4 autoantibodies over baseline (T2/T1 and T3/T1) between rAV and mRNA vaccine recipients. Secondary endpoints were the aPL autoantibody increase over baseline, the percentages of seroconversion for aPF and aPL autoantibodies (cutoff defined as 2-fold increase), the presence of aPF4 (dubious or positive), the percentage of positive PIPAA tests and the correlation between aPF4 and aPL antibodies. A moderate or strong correlation between aPF and aPL could indicate a possible shared mechanism such as a nonspecific, polyreactive autoantibody response. Finally, we explored potential correlations in the CS dataset between the post-vaccination PF4 antibody levels and the following covariates: type of vaccination, age, sex, COVID-19 experience status, number of days between vaccination and sample collection and study site (EMC/AUMC).

### 2.4. Sample Size Consideration

For the LO dataset, an effect size of 0.5 was used to calculate the difference between pre- and post-vaccination we could declare it significantly different from 0 with 80% power and 5% significance. An effect size of 0.5 roughly translates to a difference of 0.5 × SD. In this case, a sample size of 34 pairs, would allow us to detect a difference in PF4 IgG OD antibody levels of 0.0051 OD between pre- and post-vaccination. For the CS dataset, the study’s size was determined pragmatically, considering that all HCWs were included in the dataset.

### 2.5. Statistical Analysis

Continuous variables are expressed as means with standard deviation (SD) or median with interquartile range (IQR), depending on the distribution. Categorical variables are presented as frequencies with percentages. Mahalanobis distance matching was performed using the MatchIt package in R (version 4.4.0) [39], set at random order, a set seed value of 12, and no replacement. Antibody levels were compared between time points within each exposure arm using mixed modeling as implemented in GraphPad Prism 10.0.3, with Geisser-Greenhouse correction for violation of sphericity. Šídák’s multiple comparisons test was used for primary endpoints and uncorrected Fisher’s Least Significant Difference for secondary endpoints. A Spearman’s rank correlation coefficient was measured to determine the correlation between aPF4 and aPL. Seroconversion rates between vaccine groups were compared using Fisher’s exact test and the relative risk was calculated using the Koopman asymptotic score to compute (95% CI) [40,41].

We assessed correlations between PF4 autoantibody levels at T3 and the following covariates: the type of vaccine, age, sex, COVID-19 status, study site, and number of days between vaccination and venous blood sampling. We used multivariable linear regression, adjusted for different predefined confounders based on prior assumptions. These assumptions and predefined confounders are summarized in Appendix A. Linearity of the relationship between PF4 levels and age was examined using natural splines (splines package, R, version 4.3.1) and incorporated into the final model if applicable. Multiplicative interactions were explored and added to the final model if statistically significant (*p* < 0.05). Results are expressed MD with 95% CI. Statistical analyses have been performed with R (R Core Team, Vienna, Austria, 2022) and Rstudio (Rstudio Team, Boston, MA, USA, 2022) version R4.2.2 [42], and GraphPadPrism, version 10.0.3.

## 3. Results

### 3.1. Subject Characteristics

A total of 300 HCWs were excluded from the LO dataset, largely due to missing data or samples, with the remainder (*n* = 532) being split into rAV and mRNA exposure groups, which were subsequently matched for age, sex, and COVID-19 experience status. This yielded 160 matched rAV-mRNA pairs, from which the final selection was made based on sex and Mahalanobis distance. We included 775 HCWs for the cross-sectional analysis Details on numbers selected and analyzed are shown in Figure 1.

Demographics and baseline characteristics of the HCWs used in the LO and CS sets are summarized in Table 1. Overall, the HCWs were predominantly female (75.9%) and had not been previously infected with SARS-CoV-2 (75.2%). The mRNA-vaccinated HCWs were younger than the rAV vaccine recipients, both in the LO and CS datasets. The percentage of males was higher in the LO dataset, indicative of the prioritization of male pairs in the matching.

#### 3.1.1. aPF4 Levels after COVID-19 Vaccination

We did not observe any differences in aPF4 FC over baseline in the AZD1222-vaccinated individuals compared to the matched mRNA vaccine recipients at T2 or T3, with a mean FC at T2 of 0.99 versus 1.08 (MD: 0.09 [95% CI −0.24 to 0.06], *p* = 0.49) and mean FC at T3 of 0.99 versus 1.10 (MD = −0.11 [95% CI −0.31 to 0.09], *p* = 0.55) [Figure 2A–E]. Similarly, no differences were seen after a single dose of Ad26.COV2.S compared to matched controls, with a mean FC of 1.01 versus 0.99 (MD = 0.026 [95% CI −0.13 to 0.18], *p* = 0.99). The mean aPF4 levels did not change over time in any of the COVID-19 vaccine arms, with a mean (±SD) OD ratio of 0.45 (±0.26) at baseline, 0.45 (±0.28) at T2 and 0.46 (±0.24) at T3. In total, 5 (3.4%) of vaccine recipients had a dubious and 1 (0.7%) a positive PF4 ELISA at baseline. The number of dubious PF4 ELISAs decreased to 3 (2.1%) for both T2 and T3 and the number of positive PF4 ELISAs decreased to 2 (1.4%) at T2 and 0 (0.0%) at T3. These percentages were similar for the different COVID-19 vaccine arms (Figure 2A–E). The PIPAA was negative in all cases.

#### 3.1.2. aPL Antibodies after COVID-19 Vaccination

We observed small, but clinically irrelevant differences in the fold-change and mean concentration of aPL antibodies at T2 and T3 between rAV and mRNA vaccine recipients [Figure 2F–T]. For aβ2GP total Ig, the FC at T3 was a little higher in rAV vaccine recipients compared to mRNA vaccine recipients (FC 1.2 vs. 0.9, MD = 0.2 [95% CI 0.06 to 0.4], *p* = 0.007). We did not observe any differences in the FC of aβ2GP at T2 between rAV and mRNA vaccine recipients nor when looking separately at the subgroups of AZD1222 and Ad26.COV2.S. Nonetheless, the mean concentration of aβ2GP total Ig increased after the first dose of AZD1222 (Figure 2G), with a mean concentration of 5.8 at T1 versus 6.2 U/mL at T2 (MD = 0.4 [95% CI 0.1 to 0.8], *p* = 0.02) and 6.5 U/mL at T3 (MD = 0.8 (95% CI −0.1 to 1.6), *p* = 0.08). In the mRNA-1273 group, a decrease in aβ2GP total Ig was detected (Figure 2I), with a mean concentration of 4.2 U/mL at T1 versus 4.0 U/mL at T2 (MD = −0.2 [95% CI −0.5 to 0.1], *p* = 0.26) and 3.8 U/mL at T3 (MD = −0.39 [95% CI −0.7 to 0.1], *p* < 0.01).

For aCL IgG, the FC at T2 was lower in rAV vaccine recipients compared to mRNA vaccine recipients (FC 0.9 vs. 1.0, MD = −0.1 [95% CI −0.2 to −0.002, *p* = 0.05). Similarly, the FC of the Ad26.COV2.S group was lower compared to the FC of the matched mRNA vaccine recipients, with a mean FC of 0.86 versus 0.96 at T2 (MD = −0.11, [95% CI −0.20 to −0.01], *p* = 0.03). The mean antibody levels decreased after Ad26.COV2.S vaccination (Figure 2R), from 1.9 U/mL at T1 to 1.6 U/mL at T2 (MD = −0.3, [95% CI: −0.5 to −0.1], *p* < 0.01). Similarly, we observed a decrease in the mRNA-1273 group (Figure 2S), with concentrations of 2.2 U/mL at T1 versus 2.1 U/mL at T2 (MD = −0.1 [95% CI: −0.3 to 0.0], *p* = 0.17) and 1.7 U/mL at T3 (MD = −0.5 [95% CI: −0.7 to −0.3], *p* < 0.01).

We did not observe a difference in the FCs between the rAV and mRNA recipients at T2 and T3. However, when the mean anti-CL IgM levels did decrease over time in the combined mRNA group (Figure 2K) from 3.6 U/mL at T1 to 3.4 U/mL at T2 (MD = −0.1 [95% CI: −0.3 to 0.001], *p* = 0.051) and 3.3 U/mL at T3 (MD = −0.3, [95% CI: −0.6 to −0.05], *p* = 0.02). We did not observe a significant decrease when testing the mRNA-1273 and BNT162b2 subgroups separately (Figure 2N–O).

#### 3.1.3. Correlations of aPF4 with aPL Antibodies

Overall, our analysis showed no correlation between aPF4 levels and aPL (Appendix A). Nevertheless, we observed a strong consistent correlation among aPF4 levels within the same subjects across different time points (T1, T2, and T3), as well as among aPL antibodies at these time points. Additionally, there was a positive correlation between aCL IgM and aβ2GP total Ig and a positive correlation between aCL IgG and aβ2GP total Ig across all time points. When stratified by vaccine type, most of the correlations remained similar to those correlations in the main analysis (Appendix A). Notably, for the rAV-based vaccines, we observed a positive correlation between aPF4 levels and aCl IgG at T2 and T3.

### 3.2. Seroconversion

Most autoantibody levels of aPF IgG, aβ2GP total Ig, and aCL IgM and IgG remained below a 2-fold increase over baseline (T1) after the first (T2) and second dose (T3) for all COVID-19 vaccines (Figure 2). HCWs with a greater than 2-fold increase over baseline (seroconversion) at T2 or T3 for one or multiple of the four autoantibodies tested are shown individually in Table 2. Seroconversion for aPF4 IgG occurred in 3 individuals, 1/35 (2.9%) in the Ad26.COV2.S group with a T2 FC of 2.4 (Table 2 row#1), 1/36 (2.8%) after the first dose of AZD1222, with a T2 FC of 2.3 (Table 2 row#4), which normalized to 1.8 after the second dose, and 1/47 (2.1%) after the second dose of mRNA-1273, with FC of 2.5 at T3 (Table 2 row#12). In the AZD1222 group, three subjects seroconverted for multiple autoantibodies at the same time point, including the aPF4 seroconversion, which also had a 2.7-fold increase in aCL IgG over baseline (Table 2 row#4). In HCWs vaccinated with 2-dose vaccines (AZD1222, BNT162b2, and mRNA-1273) who seroconverted after the first dose, we did not observe boosting of autoantibodies after the second dose. Seroconversion against any autoantibody tested after the first vaccine dose occurred in 3 out of 35 (8.6%) in the Ad26.COV2.S group, 4 out of 36 (11.1%) for AZD1222 and 1 out of 47 (2.1%) for mRNA-1273. Overall, rAV-vaccinated individuals were significantly more likely to seroconvert at T2 for any autoantibody tested compared to mRNA-vaccine recipients.

### 3.3. Cross-Sectional Analysis of Post-Vaccination aPF4 Levels

To further explore the findings from the LO dataset, we conducted a larger cross-sectional analysis using all available post-vaccination sera. The mean (±SD) post-vaccination aPF4 levels were 0.58 (±0.43) units, with a similar proportion of dubious (*n* = 43, 5.5%) and positive ELISAs (*n* = 6, 0.8%) as observed in the LO dataset. Although most dubious ELISAs were from mRNA vaccine recipients, the comparison between rAV and mRNA vaccine recipients was not statistically significant (5/187 (0.6%) vs. 38/588 (4.9%), OR = 0.38 [95% CI: 0.096 to 1.1], *p* = 0.08). A small number of patients with dubious (4/43, 9.3%) or positive ELISAs (1/6, 16.7%) had a positive PIPAA. However, none of the reactions was blocked by the monoclonal antibody IV.3.

We explored correlations between PF4 autoantibody levels at T3 and specific covariates, including vaccine type, age, sex, COVID-19 status, study site, and days between vaccination and venous blood sampling (Figure 3). We found no significant correlation between post-vaccination aPF4 levels and the type of vaccine (mRNA vs. rAV, MD: −0.070 [95% CI −0.14 to 0.002], *p* = 0.06). Similarly, there were no significant correlations between aPF4 levels and sex, COVID-19 history, or days between vaccination and venous blood sampling. However, we observed a significant correlation between aPF4 levels and age at baseline, with an OD decrease of 0.03 [95% CI −0.06 to −0.009], *p* < 0.01) per 10 years increase in age, and a significant difference between different study sites (AUMC vs. EMC, MD = 0.12 (95% CI 0.06 to 0.19).

## 4. Discussion

This study shows that rAV and mRNA-based COVID-19 vaccines do not significantly increase aPF4 levels in the studied cohort of HCWs. Seroconversion of aPF4 occurred infrequently in both rAV and mRNA vaccine recipients, with no significant differences between the two groups. Compared to a set of control autoantibodies unrelated to VITT, aPF4 did not appear more frequently after vaccination, suggesting no specific auto-antigenic stimulus was present in any of the COVID-19 vaccines. Seroconversion for any autoantibody was more frequent in rAV vaccine recipients, and a positive correlation was found between aPF4 levels and aCL IgG in recipients of rAV, which might indicate a potential shared mechanism seen in a polyreactive antibody response. Our results suggest that, like virus infections, viral vector vaccines elicit a weak polyreactive antibody response in healthy individuals different from mRNA vaccines. However, no strong conclusions about the differences in immune response between viral vector and mRNA vaccines can be made based on this data.

Several previous studies have investigated the presence of aPF4 after rAV vaccination in individuals without VITT, with reported percentages of detectable post-vaccination aPF4 ranging from 1.2% to 8.0%, similar to our reported percentage [6,43,44,45]. Like in our study, they did not observe reproducible platelet activation upon testing the sera positive for aPF4. However, some of these studies lacked pre-vaccination measurements to exclude the possibility that these antibodies were already present before immunization [43,46]. We show that this is important because most individuals with positive aPF4 post-vaccination were already positive at baseline. Confounding also limits the validity of observational studies, which we attempted to address by matching for sex, age, and COVID-19 status. We also incorporated essential controls by studying samples from both rAV-based and mRNA vaccines and comparing aPF4 with non-specific autoantibodies. Transient increases in autoantibodies after pro-inflammatory stimuli are commonly reported, underlining the importance of including control autoantibodies to determine whether an increase in aPF4 is specific [29]. We chose aPL antibodies because they are known to be involved in polyreactive responses and they are associated with a comparable autoimmune thromboembolic disorder [32,33].

In our study, we observed a higher percentage of seroconversion of any autoantibody in rAV-based compared to mRNA-based vaccine recipients. We found one other study that reported changes in both aPL and PF4 antibodies in healthy controls after the primary vaccination series with AZD1222. The study did not detect any new aPL or PF4 autoantibody positivity nor any autoantibody response after vaccination with AZD1222 in HCWs [47]. In this study, autoantibody positivity was defined categorically using absolute cutoffs, based on clinical diagnostic criteria. We opted to use a 2-fold increase over baseline. Although this approach ignores clinical relevance, we considered it more appropriate for testing whether a vaccine-specific neoantigen was sufficiently immunogenic to induce PF4 autoantibodies. Another study specifically looked into autoantibodies after booster vaccination and found that more participants with a primary series of rAV-based vaccines had detectable aPL antibodies one day after a booster, compared to participants with mRNA-based vaccines (5/105 vs. 1/105) [48]. Importantly, the observed limited autoreactivity in these studies should not be interpreted as having implications for vaccine safety. Instead, these studies provide insights into the physiological immune response elicited by different vaccine types. Individuals predisposed to autoimmune disease commonly manifest symptoms following an immunological challenge, which can be a vaccine, but more commonly an infection or occasionally a transfusion [25]. Even in such cases, it is unlikely these challenges can induce an autoimmune disease without an underlying defect in immune tolerance.

Although mean levels of aβ2GP IgG decreased after the first and second doses of the mRNA-1273 vaccine, this study is not designed to answer the question of whether mRNA vaccines can suppress autoreactive B cells. The kinetics of aPL- and other autoantibodies after mRNA COVID-19 vaccination have been studied by others, both in patients with underlying autoimmune conditions and healthy controls, and generally found a lack of change in autoantibody levels [49,50,51,52,53,54,55]. Jaycox et al. showed stable autoantibody dynamics in both healthy individuals and patients with pre-existing underlying autoimmune disorders and contrasted these results with the frequent emergence of autoantibodies in patients with moderate to severe COVID-19 [49]. Similarly, Fiorelli et al. showed no autoreactive response changes over time in healthy subjects after BNT162b2 vaccination nor found a correlation between levels of autoantibodies and antibodies to SARS-CoV-2 in these patients [50]. These studies did not include rAV-vaccinated individuals for comparison but do suggest mRNA vaccine-induced immune responses are distinct from typical antiviral responses in their lack of humoral autoimmunity [49].

Our study aimed to provide a better understanding of aPF4 levels after COVID-19 vaccination in healthy individuals. In doing so, the strengths of this study are the use of two large biorepository studies, in which prospective and systematically sampled sera were collected. This allowed us to use rAV and mRNA vaccinated exposure groups matched for age, sex, and COVID-19 history, which potentially reduced confounding. Nonetheless, matching did not fully equalize age in the AZD1222 group compared to controls, due to a specific avoidance of this vaccine in individuals aged under 60. The concentration of autoantibodies in serum is cumulative with age, reaching a plateau in adolescence, but risk profiles may differ depending on autoantibody specificity [56]. The inclusion of the baseline measurement before vaccination and reporting the relative changes of autoantibodies is therefore useful when studying the induction of new autoantibodies by COVID-19 vaccines. Finally, the use of nonspecific control autoantibodies unrelated to VITT allowed us to determine whether any autoantibody found after vaccination is the result of antigen-specific stimulation or part of a broader autoreactive response. A limitation of this study is the lack of a more comprehensive set of autoreactive antibodies. Full autoantibody profiling could characterize the humoral response to rAV vaccines in more detail. We were also unable to determine the relevance of batch-related impurities in the induction of autoreactive antibodies, which is a possible explanation for the higher frequency of VITT, observed in AZD1222 compared to Ad26.COV2.S [4,31] and could be relevant for the development of future viral vector vaccines.

## 5. Conclusions

The precise cause of VITT has not yet been fully elucidated, as VITT is very rarely seen after COVID-19 vaccination. Heightened clinical suspicion and testing for PF4 antibodies in patients with unexplained thrombosis and thrombocytopenia, combined with the simultaneous administration of billions of rAV-based COVID-19 vaccines, may have led to a skewed impression that rAV-based vaccines are uniquely associated with VITT. The increased attention this mechanism of thrombosis has received is now leading to the retrospective discovery of more VITT-like cases, predating the COVID-19 pandemic [57,58]. Future research will provide further insight into whether VITT is exclusively a rare complication associated with rAV-based vaccines, or a previously unrecognized autoimmune syndrome with a variety of infectious and non-infectious predisposing factors, responsible for a proportion of thrombophilic disorders that have so far remained unexplained.

## Figures and Tables

**Figure 1 vaccines-11-01851-f001:**
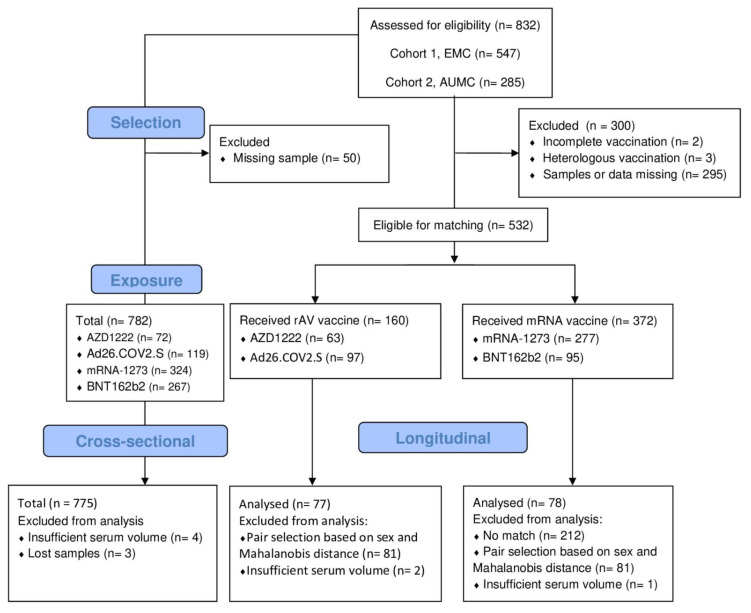
Flow diagram of participant selection for analysis. Abbreviations: EMC: Erasmus Medical Center; AUMC: Amsterdam University Medical Centers; rAV: recombinant AdenoVirus; mRNA: messenger RNA.

**Figure 2 vaccines-11-01851-f002:**
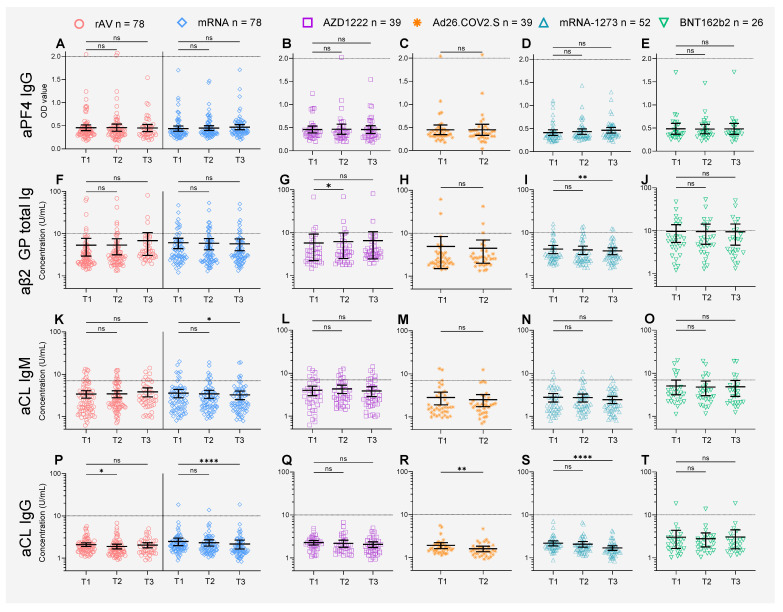
Longitudinal assessment of autoantibody responses to COVID-19 vaccines of aPF4IgG (**A**–**E**), aβ2GP (**F**–**J**), aCLIgM (**K**–**O**) and aCLIgG (**P**–**T**), in the combined rAV and mRNA groups (**A**,**F**,**K**,**P**), AZD1222 (**B**,**G**,**L**,**Q**), Ad26.COV2.S (**C**,**H**,**M**,**R**), mRNA-1273 (**D**,**I**,**N**,**S**) and BNT162b2 (**E**,**J**,**O**,**T**). Colored symbols indicate individual values, black lines are means and whiskers indicate 95% confidence intervals. The dotted lines indicate the ELISA cut-offs according to the manufacturer. Abbreviations: rAV: recombinant AdenoVirus; mRNA: messenger RNA; aPF4: Autoantibodies against Platelet Factor 4; IgG: Immunoglobulin G; Ig: Immunoglobulin; aβ2GP: anti-beta2-Glycoprotein; aCL: anti-CardioLipin; IgM: Immunoglobulin M. Mixed-modeling was used with Geisser-Greenhouse correction for violation of sphericity. Šídák’s multiple comparisons test was used for primary endpoints and uncorrected Fisher’s Least Significant Difference for secondary endpoints. ns = not significant, * = *p* < 0.05, ** = *p* < 0.01, **** = *p* < 0.0001.

**Figure 3 vaccines-11-01851-f003:**
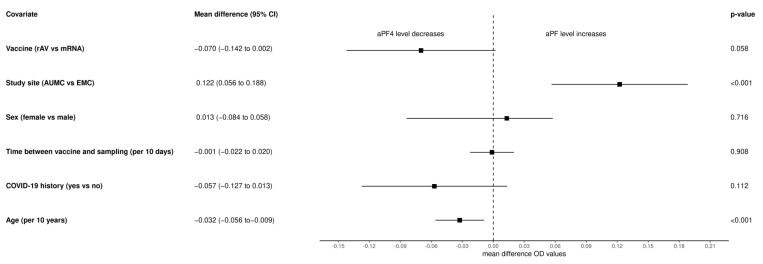
Correlations between aPF O.D values and several covariates. Forrest plot for mean differences with a 95% confidence interval for the correlation between aPF4 O.D. values and several covariates. The vaccine is the comparison of rAV-based versus mRNA-based vaccines. COVID-19 history is the comparison of yes versus no. The study site is the comparison between Amsterdam University Medical Centers vs Erasmus Medical Center. Age is expressed per 10 years from 0 years. The time between vaccine and sampling is expressed per 10 days from the day of the first COVID-19 vaccination (T1). Abbreviations: rAV: recombinant AdenoVirus; mRNA: messenger RNA; EMC: Erasmus Medical Center; AUMC: Amsterdam University Medical Center.

**Table 1 vaccines-11-01851-t001:** Demographics and baseline characteristics of HCWs used in the cross-sectional (CS) and longitudinal (LO) analysis sets.

	Dataset	rAV	mRNA	AZD1222 (rAV)	Ad26.COV2.S (rAV)	mRNA-1273 (mRNA)	BNT162b2 (mRNA)
Subjects (*n*, %)	LO	77	78	39 (25.2)	38 (25.0)	52 (33.5)	26 (16.8)
	CS	187	588	72 (9.3)	115 (14.8)	322 (41.5)	266 (34.3)
Age (mean, SD)	LO	50.4 (14.1)	43.0 (11.8)	61.6 (1.5)	39.3 (12.1)	42.8 (11.6)	43.4 (12.3)
	CS	46.4 (15.4)	46.7 (11.8)	61.0 (5.6)	37.8 (12.5)	39.5 (11.5)	41.4 (11.8)
Males (*n*, %)	LO	29 (37.2)	29 (37.2)	16 (41.0)	13 (34.2)	15 (28.8)	14 (53.8)
	CS	37 (19.8)	149 (25.3)	17 (23.6)	20 (17.4)	59 (18.3)	90 (33.8)
COVID-19 infection prior to vaccination (*n*, %)	LO	12 (15.4)	9 (11.5)	6 (15.4)	6 (15.8)	6 (11.5)	2 (7.7)
	CS	39 (20.9)	153 (26.0)	8 (11.1)	31 (27.0)	81 (25.2)	72 (27.1)

Abbreviations: rAV: recombinant AdenoVirus; mRNA: messenger RNA; SD: Standard Deviation.

**Table 2 vaccines-11-01851-t002:** Overview of HCWs in the LO dataset who seroconverted after the first (T2) or second (T3) vaccine dose.

#	Vaccine	aPF4 (FC)		aβ2GP (FC)		aCL IgG (FC)		aCL IgM (FC)	
		T2	T3	T2	T3	T2	T3	T2	T3
1	Ad26.COV2.S	**2.4**	-	1.6	-	1.0	-	0.6	-
2	Ad26.COV2.S	0.1	-	**3.4**	-	1.3	-	1.0	-
3	Ad26.COV2.S	1.2	-	1.6	-	1.3	-	**2.9**	-
4	AZD1222	**2.3**	1.8	1.5	1.2	**2.7**	1.5	1.2	1.1
5	AZD1222	0.9	0.7	0.9	**2.7**	0.8	1.7	0.9	**4.0**
6	AZD1222	-	1.1	-	**2.1**	-	1.1	-	1.4
7	AZD1222	1.3	1.3	1.3	1.9	1.1	1.3	**2.7**	1.7
8	AZD1222	1.2	1.1	1.3	1.4	0.8	0.7	**2.7**	1.5
9	AZD1222	0.9	0.6	0.9	**2.7**	0.8	1.7	0.9	**4.0**
10	AZD1222	0.7	0.8	0.9	1.4	0.8	0.7	**2.5**	1.2
11	BNT162b2	0.9	1.0	1.1	**2.7**	1.1	1.1	1.1	1.2
12	mRNA-1273	1.5	**2.5**	0.8	0.9	0.8	0.9	1.0	0.6
13	mRNA-1273	0.8	1.0	1.0	0.9	1.2	0.7	**2.2**	1.4

Abbreviations: aPF4: Autoantibodies against Platelet Factor 4; aβ2GP: anti-beta2-Glycoprotein; aCL: anti-CardioLipin. Numbers indicate fold-increase (FC) over baseline. Bold numbers indicate seroconversions (FC > 2) # = row number.

## Data Availability

The data presented in this study are available on request from the corresponding author. The data are not publicly available because our dataset contains clinical and personal data.

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
