# Peer review of "Transient Autoreactive PF4 and Antiphospholipid Antibodies in COVID-19 Vaccine Recipients"

_vaccines, 2023, doi:10.3390/vaccines11121851_

Round 1

Reviewer 1 Report

Comments and Suggestions for Authors

Authors have presented a significant research findings indicating the effect of  rAV and mRNA-based COVID-19 vaccines on the expression level of elevated aPF4 levels in healthcare workers. Overall, the structure of the research paper is highly organized. Keywords are highly effective and significant.

The abstract section summarises the research findings along with the future scope of the research work. The introduction section is a bit not organized. Also, the author is requested to introduce some more relevant studies related to this research.

Figures need significant improvement. They are not clear. All the figures are highly blurred. 

Avoid using the word outcome in between the manuscript. If you want to incorporate then add a section future outcome after the conclusion section. 

References are not cited properly. Strictly follow the instructions of the journal. 

Discussion is also not significant enough to corroborate the previous research findings.

Please check the syntax and grammatical errors in the whole manuscript.

After these suggestions, the manuscript can be considered for publication.

Comments on the Quality of English Language

Moderate editing of English language required

Author Response

[1-1] Reviewer#1 comment #1

Authors have presented a significant research findings indicating the effect of  rAV and mRNA-based COVID-19 vaccines on the expression level of elevated aPF4 levels in healthcare workers. Overall, the structure of the research paper is highly organized. Keywords are highly effective and significant.

The abstract section summarises the research findings along with the future scope of the research work. The introduction section is a bit not organized. Also, the author is requested to introduce some more relevant studies related to this research.

[R1-1] Reviewer#1 comment#1 response: Thank you for your insightful comments on our manuscript. We appreciate your acknowledgement of the significance of our research findings and the overall organization of the paper. In response to your suggestion, we have revised the introduction section, incorporating relevant studies to enhance its readability. We kindly invite you to take a look at the revised introduction.

[1-2] Reviewer#1 comment #2

Figures need significant improvement. They are not clear. All the figures are highly blurred.

[R1-2] Reviewer#1 comment#2 response: We thank you for bringing this issue to our attention. We would like to clarify that the original figures we submitted are vectorized and of high resolution. However, during the manuscript processing, changes were introduced by the journal’s formatting. In addition PDF compilation may have impacted the clarity.

In response to our comment, we have addressed this issue by reinserting the figures in the revised manuscript. Should any blurriness persist, we are more than willing to provide the high resolution figures as separate files so the journal can address this issue accordingly.

due to the changes to the original manuscripts by vaccines and compiling to pdf, the figures

[1-3] Reviewer#1 comment #3

Avoid using the word outcome in between the manuscript. If you want to incorporate then add a section future outcome after the conclusion section. 

[R1-3] Reviewer#1 comment#3 response: Thank you for your valuable suggestion regarding the use of the word "outcome" in the manuscript. Following your advice, we have replaced it with "endpoint."

[1-4] Reviewer#1 comment #4: 

Discussion is also not significant enough to corroborate the previous research findings.

[R1-4] Reviewer#1 comment#4 response: In response to your comment, we have revised and expanded certain sections of our discussion, incorporating previous research findings. We believe these revisions contribute to the overall coherence of the discussion. We kindly request you to take another look at our updated discussion section.  (see also R2-1, page 2 and R5-2, page 5).

Reviewer 2 Report

Comments and Suggestions for Authors

This study examined changes in PF4 and antiphospholipid antibodies in COVID-19 vaccine recipients. The significance, methods, and results of the study are very carefully described, with few problems. However, PF4 antibodies changed little after vaccination, whereas antiphospholipid antibodies showed a relative decrease in many cases. I felt that Discussion on this was lacking, and I believe that this point should be added.

Author Response

This study examined changes in PF4 and antiphospholipid antibodies in COVID-19 vaccine recipients. The significance, methods, and results of the study are very carefully described, with few problems. However, PF4 antibodies changed little after vaccination, whereas antiphospholipid antibodies showed a relative decrease in many cases. I felt that Discussion on this was lacking, and I believe that this point should be added.

[R2-1] Reviewer#2 comment#1 response: We appreciate your positive feedback on our manuscript and valuable comment regarding the changes in PF4 and antiphospholipid antibodies. We agree that there is a relative decrease in the antiphospholipid antibodies in many cases. However, our study is not designed to say anything about the reasons for this relative decrease and whether, for instance, mRNA vaccines can suppress autoreactive B-cells. We have incorporated the reviewer’s suggestion as follows:

“Although mean levels of aβ2GP IgG decreased after the first and second dose of the mRNA-1273 vaccine, this study is not designed to answer the question of whether mRNA vaccines can suppress autoreactive B cells. The kinetics of aPL- and other autoantibodies after mRNA COVID-19 vaccination have been studied by others, both in patients with underlying autoimmune conditions and healthy controls, and generally found a lack of change in autoantibody levels [49-55].” (Lines 414-419)

Reviewer 3 Report

Comments and Suggestions for Authors

I appreciate the opportunity to review the manuscript entitled “Transient autoreactive PF4 and antiphospholipid antibodies in COVID-19 vaccine recipients”. Developing vaccines against new viral threats is a race against time. However, at such moments it is important not to forget about possible side effects. In the wake of the COVID-19 hype, there have been many opinions and speculations about a possible increase in blood clots with the use of new COVID-19 vaccines. In their work, the authors demonstrated that vaccination with various types of COVID-19 vaccines (rAV and mRNA-based) does not increase the level of antibodies against platelet factor 4 in healthy donors. I have no comments about the manuscript in terms of experimental design, presentation, and interpretation of results. I think such a manuscript is worthy of being published in Vaccines. However, I have a number of minor comments regarding the layout of the text of the manuscript. You can read these comments below.

1. Please cite literary sources in accordance with the journal's rules. From "Instructions for Authors", Vaccines, MDPI. In the text, reference numbers should be placed in square brackets [ ], and placed before the punctuation; for example [1], [1–3] or [1,3]. For embedded citations in the text with pagination, use both parentheses and brackets to indicate the reference number and page numbers; for example [5] (p. 10). or [6] (pp. 101–105).

2. Line 277, 279, 288, 293, 296, 309. Capitalize the word Figure.

3. Line 290. You missed the period at the end of the sentence and paragraph.

4. Line 311, 312, 313, 315, 317. Capitalize the word Table.

5. Figure 3. Can you please make all the labels on the graph a little larger? In this version of the Figure 3, the labels on the graph are quite difficult to read.

Author Response

I appreciate the opportunity to review the manuscript entitled “Transient autoreactive PF4 and antiphospholipid antibodies in COVID-19 vaccine recipients”. Developing vaccines against new viral threats is a race against time. However, at such moments it is important not to forget about possible side effects. In the wake of the COVID-19 hype, there have been many opinions and speculations about a possible increase in blood clots with the use of new COVID-19 vaccines. In their work, the authors demonstrated that vaccination with various types of COVID-19 vaccines (rAV and mRNA-based) does not increase the level of antibodies against platelet factor 4 in healthy donors. I have no comments about the manuscript in terms of experimental design, presentation, and interpretation of results. I think such a manuscript is worthy of being published in Vaccines. However, I have a number of minor comments regarding the layout of the text of the manuscript. You can read these comments below.

  1. Please cite literary sources in accordance with the journal's rules. From "Instructions for Authors", Vaccines, MDPI.In the text, reference numbers should be placed in square brackets [ ], and placed before the punctuation; for example [1], [1–3] or [1,3]. For embedded citations in the text with pagination, use both parentheses and brackets to indicate the reference number and page numbers; for example [5] (p. 10). or [6] (pp. 101–105).
  2. Line 277, 279, 288, 293, 296, 309. Capitalize the word 
  3. Line 290. You missed the period at the end of the sentence and paragraph.
  4. Line 311, 312, 313, 315, 317. Capitalize the word Table.

[R3-1] Reviewer#3 comment#1 response: We would like to thank the reviewer for their time and thoughtful review of our manuscript. We appreciate your positive feedback on our design, presentation and interpretation. We have carefully revised the manuscript, ensuring adherence to the citation style of Vaccines. In addition, we have addressed the other minor layout issues as well.

[3-2] Reviewer#3 comment #2: 

Figure 3. Can you please make all the labels on the graph a little larger? In this version of the Figure 3, the labels on the graph are quite difficult to read.

[R3-2] Reviewer#3 comment#2 response: Thank you for your valuable suggestion. Our original manuscript was changed by vaccines to facilitated the peer review process. During these editorial changes, the figures have become smaller and more blurry. In our revised manuscript, we have inserted the figures and made them bigger. We hope this will enhance the readability of Figure 3.  If the figures are still blurry or unreadably, we are more than happy to provide our high resolution figures as separated files so the journal can address this issue accordingly.

Reviewer 4 Report

Comments and Suggestions for Authors

The authors provide a comprehensive study on the analysis of the findings related to vaccine-induced immune thrombotic thrombocytopenia (VITT) in the context of recombinant adenovirus (rAV) and mRNA-based COVID-19 vaccines. In the paper, four vaccination groups are employed to compare the antibody fold change for autoantibodies against Platelet Factor 4 (aPF4) and antiphospholipid antibodies (aPL). The data shown in the paper clearly revealed no significant differences in aPF4 fold changes after the first and second doses of rAV-based vaccines compared to mRNA vaccines. The study concludes that both rAV and mRNA-based COVID-19 vaccines do not markedly increase aPF4 levels in healthy individuals. In general, the work offers a thorough interpretation of the study's findings, discusses the implications in the context of existing literature, and identifies areas for future research. Minor suggestions are listed below:

1.     What is the rationale for administering a single dose of Ad26.COV2.S as opposed to the two doses given in the other three groups?

2.     On Page 264, “Ig; Immunoglobuline” is “Ig: Immunoglobuline;”

3.     On Page 268, **** = P< 0.00 can be corrected to **** = P< 0.0001.

Author Response

The authors provide a comprehensive study on the analysis of the findings related to vaccine-induced immune thrombotic thrombocytopenia (VITT) in the context of recombinant adenovirus (rAV) and mRNA-based COVID-19 vaccines. In the paper, four vaccination groups are employed to compare the antibody fold change for autoantibodies against Platelet Factor 4 (aPF4) and antiphospholipid antibodies (aPL). The data shown in the paper clearly revealed no significant differences in aPF4 fold changes after the first and second doses of rAV-based vaccines compared to mRNA vaccines. The study concludes that both rAV and mRNA-based COVID-19 vaccines do not markedly increase aPF4 levels in healthy individuals. In general, the work offers a thorough interpretation of the study's findings, discusses the implications in the context of existing literature, and identifies areas for future research. Minor suggestions are listed below:

What is the rationale for administering a single dose of Ad26.COV2.S as opposed to the two doses given in the other three groups?

[R4-1] Reviewer#4 comment#1 response: We appreciate your thorough review and positive feedback on our manuscript. We acknowledge the importance of clarifying the single dose of  Ad26.COV2.S. As Ad26.COV2.S is licensed as a single-injection vaccine, we have now included a reference to this in the methods section of the revised manuscript:

“Of note, HCWs were aware of the type they would receive before their scheduled vaccina-tion appointment, which means we cannot exclude self-selection. Vaccines were dosed in accordance with the manufacturer’s instructions at the time of their introduction, which specified two doses, except for Ad26.COV.2.S, which is licensed as a single-dose vaccine [37].” (Lines 119-123)

[4-2] Reviewer#4 comment #2

On Page 264, “Ig; Immunoglobuline” is “Ig: Immunoglobuline;”

[R4-2] Reviewer#4 comment#2 response: Thank you for identifying the typo on line 264. We appreciate your observation and we have changed the error as you suggested and changed immunoglobuline into immunoglobulin.

[4-3] Reviewer#4 comment #3

On Page 268, **** = P< 0.00 can be corrected to **** = P< 0.0001.

[R4-3] Reviewer#4 comment#3 response: We also appreciate your observation regarding the correction needed on line 268. The notation  *** = P< 0.00 has been updated to **** = P< 0.0001.

Reviewer 5 Report

Comments and Suggestions for Authors

Review20

The authors investigated the effects of COVID-19 vaccines on autoreactive PF4 and antiphospholipid antibodies. Their study revealed that both rAV and mRNA-based COVID-19 vaccines did not significantly increase aPF4 antibody levels in the studied cohort of healthcare workers. Seroconversion of aPF4 occurred infrequently in both rAV and mRNA vaccine recipients, with no significant differences between the two groups. The authors concluded that COVID-19 vaccines did not appear to increase the risk of developing blood clotting disorders associated with aPF4 and antiphospholipid antibodies. It is a valuable study to address a widely discussed vaccine safety issue, the relationship between COVID-19 rAV and mRNA vaccines and vaccine-induced immune thrombotic thrombocytopenia (VITT).

Here are some questions and suggestions.

1. The authors should discuss more mechanisms of vaccine-induced immune thrombotic thrombocytopenia (VITT)  beyond aPF4 antibody in the introduction and discussion.

2. The authors obtained clear results of the measured autoreactive . However, considering the very low frequency of VITT, the authors have to provide more discussion on the significance of the measured autoreactive antibodies, especially the difference between mRNA vaccine and rAV vaccines. 

BTW, I did not find the supplementary documents of this article. The uploaded file seems another version of the manuscript, not supplementary data.

Author Response

The authors investigated the effects of COVID-19 vaccines on autoreactive PF4 and antiphospholipid antibodies. Their study revealed that both rAV and mRNA-based COVID-19 vaccines did not significantly increase aPF4 antibody levels in the studied cohort of healthcare workers. Seroconversion of aPF4 occurred infrequently in both rAV and mRNA vaccine recipients, with no significant differences between the two groups. The authors concluded that COVID-19 vaccines did not appear to increase the risk of developing blood clotting disorders associated with aPF4 and antiphospholipid antibodies. It is a valuable study to address a widely discussed vaccine safety issue, the relationship between COVID-19 rAV and mRNA vaccines and vaccine-induced immune thrombotic thrombocytopenia (VITT).

 Here are some questions and suggestions.

The authors should discuss more mechanisms of vaccine-induced immune thrombotic thrombocytopenia (VITT)  beyond aPF4 antibody in the introduction and discussion.

[R5-1] Reviewer#5 comment#1 response: We appreciate your evaluation of our manuscript and we thank the reviewer for their valuable suggestion regarding the discussion of additional mechanisms of VITT beyond aPF4 antibodies. While we acknowledge the existence of multiple mechanisms, we have chosen to maintain our focus on PF4 antibodies in our manuscript. We believe this approach aligns with the main objectives of our study and have chosen to not discuss more mechanisms of VITT in the introduction and discussion.

[5-2] Reviewer#5 comment #2

The authors obtained clear results of the measured autoreactive. However, considering the very low frequency of VITT, the authors have to provide more discussion on the significance of the measured autoreactive antibodies, especially the difference between mRNA vaccine and rAV vaccines. 

[R5-2] Reviewer#5 comment#2 response: we thank you for highlighting the need for more discussion on the significance of the measured autoreactive autoantibodies. We have addressed this concern by providing additional context in the discussion section as follows:

“Another study specifically looked into autoantibodies after booster vaccination and found that more participants with a primary series of rAV-based vaccines had detectable aPL antibodies one day after a booster, compared to participants with mRNA-based vaccines (5/105 vs 1/105) [48]. Importantly, the observed limited autoreactivity in these studies should not be interpreted as having implications for vaccine safety. Instead, these studies provide insights into the physiological immune response elicited by different vaccine types. Individuals predisposed to autoimmune disease commonly manifest symptoms following an immunological challenge, which can be a vaccine, but more commonly an infection or occasionally a transfusion [25]. Even in such cases, it is unlikely these challenges can induce an autoimmune disease without an underlying defect in immune tolerance.

Although mean levels of aβ2GP IgG decreased after the first and second dose of the mRNA-1273 vaccine, this study is not designed to answer the question of whether mRNA vaccines can suppress autoreactive B cells. The kinetics of aPL- and other autoantibodies after mRNA COVID-19 vaccination have been studied by others, both in patients with underlying autoimmune conditions and healthy controls, and generally found a lack of change in autoantibody levels [49-55].” (lines 403-416)

[5-3] Reviewer#5 comment #3

BTW, I did not find the supplementary documents of this article. The uploaded file seems another version of the manuscript, not supplementary data.

[R5-3] Reviewer#5 comment#3 response: We have now added the supplementary documents and apologize for any inconvenience caused by the initial file upload.